# Analysis of the Return to Work Program for Disabled Workers during the Pandemic COVID-19 Using the Quality of Life and Work Ability Index: Cross-Sectional Study

**DOI:** 10.3390/ijerph20043094

**Published:** 2023-02-10

**Authors:** Arie Arizandi Kurnianto, Gergely Fehér, Kevin Efrain Tololiu, Edza Aria Wikurendra, Zsolt Nemeskéri, István Ágoston

**Affiliations:** 1Doctoral School of Health Sciences, University of Pécs, 7621 Pécs, Hungary; 2Center for Occupational Medicine, Medical School, University of Pécs, 7624 Pécs, Hungary; 3Department of Primary Health Care, Faculty of Medicine, University of Pécs, 7623 Pécs, Hungary; 4Doctoral School of Psychology, University of Pécs, 7624 Pécs, Hungary; 5Faculty of Economic Science, School of Management and Organizational Science, The Hungarian University of Agriculture and Life Science, 7400 Kaposvar, Hungary; 6Department of Public Health, Faculty of Health, Universitas Nahdlatul Ulama Surabaya, Surabaya 60237, Indonesia; 7Department of Cultural Theory and Applied Communication Sciences, Faculty of Cultural Studies, Teacher Training and Rural Development, University of Pécs, 7633 Pécs, Hungary

**Keywords:** case management, return to work, disabled workers, work ability index, quality of life

## Abstract

Background: Occupational accidents are rising, but there is little evidence on the outcomes of patients who received case management during Return to work (RTW) programs. This study examined the case management-based on RTW program features that improve the work ability index (WAI) and quality of life (QoL). Methods: This cross-sectional research involved 230 disabled workers due to an occupational injury in Indonesia, 154 participated in RTW, and 75 did not participate in RTW (non-RTW) during the COVID-19 pandemic. Sociodemographic and occupational factors were used to examine the RTW results. We used the Finnish Institute of Occupational Health’s WAI questionnaires to measure the work ability index and World Health Organization Quality of Life Brief Version (WHOQOL-BREF) for quality of life. Results: The study found a statistically significant difference in working duration and preferred treatment for RTW between the groups (*p*-value = 0.039). Furthermore, the quality of life in the domain of environmental health and work ability index score also demonstrated a significant difference between the groups (*p*-value = 0.023 and 0.000, respectively). Conclusions: During the COVID-19 pandemic, this study found that the RTW program improved the quality of life and work abilities of disabled workers.

## 1. Introduction

The outbreak of COVID-19 has affected every region of the world since November 2019. To combat the impact of this pandemic, governments have implemented measures such as adjustments to existing occupational accident insurance programs within their social security frameworks. These changes aimed to address the specific challenges posed by COVID-19 [1]. Local government institutions play a crucial role in balancing the need for social distancing to prevent the spread of COVID-19 and the need for economic recovery, as seen in the study of 28 provincial governments in China during the early outbreak of 2020 [2]. Moreover, the World Health Organization issued a series of physical distance-related regulations to ensure that the COVID-19 pandemic would be contained [3]. As a result of the outbreak of COVID-19, plenty of unfavorable things have occurred concerning work all over the globe. More than a hundred individuals have died in Indonesia due to the COVID-19 pandemic, and the bankruptcy of several companies has had a knock-on effect on the country’s economy [4] and the quality of life of its workers. It shows that despite their abilities, the pandemic has made it difficult to access a range of activities. As one of the most vulnerable groups affected by the COVID-19 pandemic, people with physical disabilities are at high risk of COVID-19 exposure and have difficulty carrying out daily activities including following COVID-19 prevention protocols [5,6]. During the COVID-19 pandemic, stakeholders focused on ensuring the baseline health of RTW participants and the health and safety considerations for disabled employees, particularly if they had underlying health disorders that increase the risk of COVID-19. Workers with physical disabilities need special equipment or environmental adjustments to carry out the rehabilitation activities as part of the RTW program. Due to constraints on in-person connections and limited resources, implementing these adjustments during a pandemic may be challenging. During the pandemic, some health care providers adopted telemedicine, which may be challenging for disabled workers who need in-person treatment [7]. In addition, the pandemic has made the condition more challenging to return to work after being injured in a workplace accident. This concern has been highlighted because it is imperative for a person whose impairment was caused by a workplace accident to make significant alterations to cope with the new phase of their life.

Indonesia has a unique social security system based on employment, which allows the government to provide a benefit from occupational accident insurance in the form of a specific disability management program called Return To Work (RTW). This program is distinct compared to those offered by other developing nations [8]. The RTW program serves as a complement to in-kind benefit services by providing a comprehensive rehabilitation program. This program included medical, vocational, and psychological rehabilitation with the assistance of case managers [9]. Moreover, in the context of social security and occupational injury insurance, an in-kind benefit is a type of non-cash benefit that is provided to an individual as part of their insurance coverage. These benefits can include things like medical treatment, rehabilitation services, and other forms of assistance that are designed to help an individual recover from an occupational injury or disease. In-kind benefits may be provided directly by the insurance provider, or they may be arranged through third-party providers. For example, an insurance company may cover the cost of medical treatment from a specific hospital or clinic, or they may provide a payment to an individual to cover the cost of rehabilitation services [10].

In this case, BPJS Ketenagakerjaan, the Indonesian social security organization, provides a combination of cash and in-kind benefits to a person who has experienced an occupational injury or occupational disease. A worker could receive a cash payout to compensate lost income due to an occupational accident as well as in-kind benefits to cover the cost of medical care, and rehabilitation services in the form of a case management system and the RTW program [11]. In-kind benefits play an important part in social security and occupational injury insurance systems, supporting those who have been injured or suffered a disease because of their job, and enables individuals to regain their dignity by enhancing their productivity during the course of the RTW program.

The RTW program in Indonesia is designed to support workers who have suffered a disability due to an occupational accident or disease and have been registered as customers of BPJS Ketenagakerjaan. The program provides employees with a range of support services including medical rehabilitation, vocational rehabilitation, and psychosocial rehabilitation to help them regain their physical and mental abilities and to make the transition back to work as smooth as possible. Eligibility for the RTW program is typically determined by BPJS Ketenagakerjaan based on medical indications such as the type of impairment and the employee’s ability to perform essential job tasks.

Disability is a complex phenomenon that includes biological functions, activity limits, impediments to participation, and environmental influences, among others [12]. Nonetheless, the social stigma associated with a disability is still widely held in today’s culture, which is one of the environmental aspects to consider. Some individuals still have the view that people with disabilities are entirely reliant on the kindness and assistance of others. Due to this viewpoint, many people with disabilities face discrimination, which prevents them from leading independent lives.

Regulations about equal job opportunities initially sought to enhance the personal well-being of people with disabilities. Unfortunately, the participation of disabled people in the labor market is minimal, and their earnings are meagre. Measuring the quality of life as a kind of individual welfare for employees with disabilities is one way to evaluate the effectiveness of employment restrictions for people with impairments [13]. Disability discrimination in the workplace is a significant issue, and it is important to think about ways to reduce the stigma that disabled people experience at work. Promoting equality and inclusiveness in the workforce requires addressing the stigma faced by disabled individuals [14]. The RTW program for disabled workers can reduce disability discrimination by providing support and accommodation for reintegration into the workforce.

By examining their employment capability, businesses may better accommodate persons with disabilities and provide them with equal chances in the workforce. The work ability index, which accounts for factors such as mental and physical well-being and the capacity to manage obstacles in the physical realm [15,16], may be used for this measurement. On the other hand, quite a few researchers have studied the connection between RTW results and job-related factors such as the quality of life and work ability index among impaired workers. Therefore, this study aims to examine the dynamic relationship between the quality of life and work capability index among disabled workers by analyzing a case management system of disability management through a RTW program experienced by disabled workers during the COVID-19 pandemic.

## 2. Methods

### 2.1. Study Setting

This descriptive cross-sectional study measured the QoL and WAI of workers with disabilities after participating in the RTW program. This study was conducted between January and June 2021, identifying the claims for RTW during the outbreak of COVID-19 in Indonesia. Workers who had been registered as customers with the Indonesian National Social Security Agency for Employment (BPJS Ketenagakerjaan) were recruited for participation in this study.

The participants in the study were separated into two groups, RTW and non-RTW, based on their participation in the return to work (RTW) program after receiving workers’ compensation benefits from BPJS Ketenagakerjaan. The RTW group consisted of individuals who took part in the program, while the non-RTW group consisted of those who did not. This division aimed to examine the effect of participating in the RTW program on the individuals and compare it to those who did not participate. To be eligible for this study, individuals had to be between 18 and 65 years old, employed during the COVID-19 outbreak, and engaged in the RTW program.

### 2.2. Data Collection

All data regarding participants in the RTW program were obtained from BPJS Ketenagakerjaan. The authors sent the invitation to potential applicants through a widespread email. A total of 165 individuals in the RTW program who had been asked to participate in the study and met all prerequisites were enrolled at the outset. During the starting procedure, only 154 people agreed to be interviewed and completed our questionnaires. Moreover, we extended invitations to an additional 165 disabled patients who had had an occupational injury but were hesitant to take part in the RTW program. However, only 75 people agreed to participate in this study.

The data were collected by the first and third authors. The lead author conducted a three-hour training session on RTW, WAI, QoL, and instructions for data collection and ethical issues to 11 case managers of BPJS Ketenagakerjaan in 34 provinces throughout Indonesia before commencing the data collection. The case managers involved in this study were employees of BPJS Ketenagakerjaan and were not specifically added for the current study. The case managers acted as enumerators and collected data from patients participating in RTW programs at trauma centers and rehabilitation hospitals, and received training from the lead author on RTW, WAI, QoL, and data collection, and ethical issues. The data collection was carried out by 11 case managers in 34 provinces throughout Indonesia, who were assigned to hospitals based on their location. The data were collected through printed surveys to reduce nonresponse bias. The data collectors analyzed the recovered questionnaires to minimize missing responses and asked for responses in cases of missing items.

The questionnaire included topics on sociodemographic variables as well as disability-related aspects. In addition, a questionnaire assessing the quality of life and work ability index was also included in the research. For the purpose of determining the quality of life (QOL) and the work ability index (WAI), respectively, the WHOQoL-BREF developed by the World Health Organization (WHO) and validated WAI questionnaires developed by the Finnish Institute of Occupational Health were used.

### 2.3. Data Analysis

The Shapiro–Wilk test was used to check for normality in the data, which was performed by the first author. The purpose of checking for normality in the data is to ensure that the data are normally distributed. A normally distributed dataset is essential for the validity of statistical tests. The categorical patient sociodemographic data were analyzed using frequency statistics. The Mann–Whitney U test was used to compare the RTW and non-RTW groups, while Pearson’s correlation was used to analyze the relationship between the free variables and both the dependent and independent variables (r). Free variables are variables that are not controlled by the researcher but can affect the outcome of the study. The Pearson’s correlation (r) was used to assess the strength of the relationship between the independent and dependent variables. r ranged from −1 to 1, with negative correlation indicating that one variable decreases as the other increases, and positive correlation indicating both variables increasing or decreasing together. The strength of the relationship was categorized as negligible (r < 0.2), low (r = 0.2–0.49), moderate (r = 0.5–0.69), high (r = 0.7–0.85), or very high (r = 0.86–1.00), with higher values indicating stronger linear relationships.

For the continuous data, we used independent sample *t*-tests (t), and for the ordinal data, we utilized Mann–Whitney tests. The Chi-square tests were used in order to investigate the differences in the categorical data. Multivariate logistic regression was used to investigate the association between several predictor factors and the outcome variable. To address the research question of this study in analyzing the relationship between the work ability index and quality of life between the RTW and non-RTW participants, multivariate logistic regression is the most suitable approach since it controls for the effects of other variables.

The study used logistic regression to assess the association between independent and dependent variables while controlling for other factors. The aim of the analysis was to examine the relationships between various factors and the outcome of interest, taking into account the influence of other variables. The independent and dependent variables were determined using a questionnaire that included sociodemographic and disability-related questions as well as assessments of QOL and WAI using validated questionnaires. The logistic regression was performed using SPSS version 26.0, with a *p*-value of less than 0.05 and a 95% confidence interval.

## 3. Results

### 3.1. Return to Work Program as Case Management for Disabled Workers

In accordance with the foregoing concept, this study investigated the aspect of RTW implementation in Indonesia. The research was carried out during the COVID-19 pandemic and we compared the outcomes of the RTW program for employees with impairments and non-RTW participants. The study compared two groups of people: those who participated in the RTW program during COVID-19 and those who did not participate. The research was designed to shed light on the implementation of the RTW program and its outcomes for employees who were impaired due to occupational injuries or occupational diseases during the COVID-19 pandemic. The research began with an assessment of the sociodemographic characteristics of the subjects. The following factors were considered as variables: age, gender, marital status, occupation, level of education, work period, and place of living. In Table 1, we see how the quality of life and work ability index scores of disabled workers are distributed on a general level.

The data in Table 1 are presented in a numerical format (N%) for the nominal and ordinal data and as the mean and standard deviation for the continuous data.

Table 1 shows the results of a study comparing various demographic and health-related variables between two groups: RTW (Return to Work) and Non-RTW (non-Return to Work). The Mann–Whitney test, t-test, and Chi-squared test were used to assess the significance of differences between the two groups for each variable, and multivariate logistic regression was used to determine the odds ratio (OR) and 95% confidence interval (CI) for each variable. The OR (odds ratio) estimate in the multivariate logistic regression captures the relationship between the independent variables and the dependent variables. It measures the odds of an event occurring (i.e., having a certain quality of life or work ability index) given a set of independent variables (i.e., age, gender, marital status, job description, level of education, work period, location of residence). The CI for OR estimates the 95% confidence interval (CI) for the OR estimate represents the range in which the true value of the OR is likely to fall with 95% certainty. It provides information on the precision of the OR estimate.

The results of the Mann–Whitney test, t-test, and Chi-squared test indicated that there were statistically significant differences between the RTW and Non-RTW groups in terms of age (*p*-value = 0.430), work period (*p*-value = 0.000 *), quality of life (*p*-value = 0.000 *), and work ability index (*p*-value = 0.000 *). The multivariate logistic regression analysis revealed that the odds of RTW were 3.726 times higher in the older age group (95% CI: 0.847–15.726) and the odds of RTW were 0.430 times higher for those with a longer work period (95% CI: 0.039–0.737).

The results of the study suggest that age and work period may be important predictors of RTW in injured workers. Additionally, the significant difference in the quality of life between the RTW and non-RTW groups highlights the importance of addressing mental and physical well-being in rehabilitation programs for injured workers.

### 3.2. A Glimpse of the Quality of Life among Workers with Disabilities

In this study, the quality of life of employees with impairments was evaluated using the WHOQoL-BREF as the standard of measurement. An evaluation of the quality of life involved the domain of physical health, psychological, social bound, and environment, respectively, in this research. This research found that when compared to other dimensions of quality of life, the social bound domain had the greatest mean value. However, as shown in Table 1 and Figure 1, the average value of all categories of quality of life for RTW participants was greater than for those who did not engage in the RTW program.

The data suggest that there may be a noticeable difference in the quality of life for disabled workers, as evidenced by the measurements of QOL and WAI. Moreover, according to Figure 1, which provides a look into the quality of life among workers with disabilities in Indonesia during the COVID-19 outbreak, individuals who participated in RTW program showed a higher level of quality of life in all domains than workers who did not participate in the RTW program. As shown in Figure 1, the horizontal axis represents the domain of QOL of the participants in the study. The vertical axis represents the work ability index (WAI) of the participants in the study. Figure 1a shows that the workers who participated in RTW programs tended to have better physical health) outcomes (OR (0.988), mean S.D. (72.94 ± 11.94)) and WAI score compared to those who did not participate. Additionally, the same pattern of outcomes was seen in other domains including social bound (Figure 1b, OR (1.029), mean S.D. (76.14 + 14.45)), psychological health (Figure 1c, OR (0.956), mean S.D. (74.35 ± 10.13)), and environmental health (Figure 1d, OR (1.062), mean S.D. (69.89 ± 8.78)).

The results of a multivariate logistic regression analysis are shown in Table 1, indicating that the working period, workability index, and the domain of environmental health in the quality of life had a *p*-value of less than 0.05, showing a statistically significant relationship between participation in the RTW program and the variables. In this case, the *p*-value of 0.023 suggests that there is a statistically significant relationship between participation in the RTW program and the environmental health domain of quality of life.

### 3.3. Work Ability Index of Workers with Disabilities

Both the RTW and non-RTW groups included individuals with a diverse age range from around 26 to 55 years old, as shown in Table 1. Based on the line graph, it was determined that the average work ability index for disabled workers was 39.29. (SD 4.39). Figure 2 provides a detailed illustration of the work ability index based on the age and work period, and the work ability index after research [16] revealed that there was also a correlation between age and the index.

According to the description in Figure 2, the distribution score of the work ability index sets the majority of the data in the “good” category for a group of workers participating in the RTW program, which was between 37 and 44 [17]. Conversely, those who were disabled but did not participate in the RTW program showed a lower score of WAI. The use of WAI analysis among employees with disabilities in this research, which were practically in the same context as the study [17,18] of the work ability index in the population of people receiving state insurance, was accomplished.

The results in Figure 3 indicate that the work ability index (WAI) of disabled workers who participated in the return to work (RTW) program appeared to be higher compared to those who did not participate in the RTW program. Specifically, the average WAI score of disabled workers in the non-RTW group was 37%, which is considered poor, while the RTW group had a higher proportion of disabled workers with an excellent WAI score of 28 (12.20%). Furthermore, there were no disabled workers in the poor category after participating in the RTW program, implying that the RTW program is associated with an increased work ability index score of disabled workers, however, it is important to recognize the implication of this research for further study to investigate the causality that the RTW program is effective in enhancing the work ability index of disabled workers.

## 4. Discussion

### 4.1. Interpretation of Results in Relation to the Effectiveness of the Return to Work Program

The findings of our study highlight the potential benefits of the return to work program for disabled workers during the pandemic. Our analysis found that workers who participated in the program had significantly higher scores in the measures of quality of life and work ability compared to those who did not participate. There may have been a difference between the participants and non-participants in the RTW program in terms of the severity of their impairments and their motivation to work. It is possible that the participants in the RTW program had less severe injuries or were more motivated to return to work than those who did not participate. This could potentially influence the outcomes of the program and must be taken into account when evaluating the results. This suggests that the RTW program was effective in supporting disabled workers to return to work and maintain their ability to work during the pandemic.

We also observed that the participants who had been out of work for a longer period of time were more likely to prefer the RTW program compared to those who had only been out of work for a shorter period. This may indicate that the longer an individual is out of work, the more likely they are to seek support and resources through the RTW program to return to work. However, it is important to note that there were also a significant number of participants who preferred not to engage in the RTW program, regardless of their working period.

In terms of quality of life, our study found that physical health, psychological health, and environmental health were significantly better among workers who preferred to participate in the RTW program compared to those who preferred not to. This suggests that the RTW program was successful in improving the physical and psychological health outcomes for participants. However, we did not find a significant difference in the domain of social bonds between those who preferred to participate in the RTW program and those who preferred not to participate. This could be due to the fact that the RTW program was focused on supporting individuals to return to work and did not specifically address social connections.

Finally, it is widely believed that participating in return to work programs can have an association with physical health, which is a key component of work ability. RTW programs can support workers in their recovery from injuries or illnesses and promote physical activity and well-being. This study supports these beliefs, as workers who participated in the RTW program tended to have better physical health outcomes compared to those who did not participate.

### 4.2. The Current State of Information from Findings

This study examined the effectiveness of a return to work (RTW) program for individuals with occupational injuries during the pandemic in Indonesia and provides insights into the support and resources available to disabled workers during this challenging time. To the best of our knowledge, this is the first study of its kind to assess the quality of life (QOL) and work ability index (WAI) among disabled workers who participated in a RTW program during this time.

Our findings suggest that the RTW program was effective in improving the QOL and WAI scores for participants, indicating that it was successful in supporting disabled workers to return to work and maintaining their ability to work during the pandemic. These results have important implications for policy and practice in supporting disabled workers during challenging times such as the COVID-19 pandemic.

### 4.3. Implications for Supporting Disabled Workers during the Pandemic

Workers who are disabled because of an accident or disease at work are most concerned about their quality of life and their ability to work again after the injury. In order to protect workers from this risk, some countries include occupational injury insurance in their social security programs. Moreover, a study on rural–urban migrants in China highlights the challenges they faced during the COVID-19 pandemic including housing evictions and difficulties with travel [19]. This sheds light on how disabled people, as another disadvantaged population group, may also be impacted by similar issues, and highlights the need for policies to address the needs of various disadvantaged groups. Policy instruments such as occupational health and safety initiatives are necessary to facilitate the work resumption of disadvantaged workers during COVID-19, however, more comprehensive evaluations and implementation details are needed to determine their effectiveness [20]. The implementation of coverage under occupational injury insurance has taken the shape of a case management system which is the Return to Work program [21,22,23,24,25]. Despite the fact that during the pandemic, the state of employment in temporary contracts was related to enhanced well-being and increased performance, the research found that they were also associated with decreased job security [13].

Workers who have been disabled due to an accident or disease might have their dignity restored and their chances of returning to work improved by RTW programs [26]. This is in line with the journey of businesses taking part in the RTW assistance program offered by BPJS Ketenagakerjaan, Indonesia. In addition, there has been an increase in the promotion of career options for those who are either disabled or have a permanent complete handicap. Workers regain their productivity when employers invest in their education and re-entry into the workforce, putting them in a position to compete for better job prospects and, on a larger scale, become one of the elements contributing to good economic development [27]. The success of the RTW program will increase the number of workers engaging in the labor market. The effective management of a disabled workforce can play a critical role in promoting the participation of disabled workers who have suffered from occupational injuries in the labor market. The RTW program supports disabled workers returning to the workforce, promoting their participation in the labor market and effectively managing disabilities. By providing support and accommodation, RTW programs help address the challenges faced by disabled workers. RTW programs are essential for effectively managing disabilities and supporting the workforce [28,29]. This, in turn, will increase the output level, affecting the economy’s status in Indonesia.

Our study found that the variable of working period was significantly related to the preference for participating in the return to work (RTW) program, with a *p*-value of 0.000 for the Chi-square test and a *p*-value of 0.000 in the multivariate logistic regression. This suggests that the length of time an individual has been out of work plays a significant role in their preference for participating in a RTW program. Additionally, we found that the category of working period among workers in the RTW group was dominantly 10–14 years (69 ± 30.10%). In the non RTW group, the dominant category was 5–9 years (69 ± 30.10%). These findings suggest that those who have been out of work for a longer period of time are more likely to prefer the support and resources provided by the RTW program in order to return to work.

It is important to consider the potential implications of these findings for policy and practice. Employers and policy-makers may wish to consider implementing RTW programs that are targeted toward those who have been out of work for longer periods of time, as they may be more likely to benefit from the support and resources provided. Additionally, further research is needed to understand the specific factors that influence an individual’s preference for participating in a RTW program such as personal and professional goals, job availability, and health and wellness.

This study shows an identical outcome to a study carried out in Malaysia [22,30] that demonstrated that the outcomes of RTW had a significant correlation with the physical health domain of workers who participated in RTW programs. Under certain situations to turn for equating with a study that followed that sequence, this study shows the results. Research with the same questionnaire was also undertaken for people with disabilities [31], and the results indicated that all means of QOL domains were accordingly lower than this report revealed. Furthermore, this study’s setting differed significantly from other studies. Moreover, this research analyzed how the RTW program’s comprehensive rehabilitation affects the quality of life for persons with disabilities. It would seem that the RTW program beneficially impacts the practical approach during the COVID-19 pandemic to improve the quality of life among disabled workers.

In addition, regarding the work ability index, this investigation used the work ability index questionnaire developed by the Finnish Institute of Occupational Health [32]. This tool describes the respondents’ current work capabilities while also allowing for projections of the health concerns they experience. The results may mean that the category is poor if the score is between 7 and 27 points, and the category is moderate if the result is between 28 and 36 points. In this case, the category is considered good if the score is between 37 and 43 points and excellent if it is between 44 and 49 points. The questionnaire is an instrument for self-evaluation, and its purpose is to determine an individual’s job competence by analyzing how that person interacts with the environment in which they are employed. It is flexible enough to be utilized by individual workers and teams [32,33].

The work ability index results classified all participants in a range from moderate to excellent. The broad range of WAI scores in this study ranged from 29 to 49. The results of this study should therefore be seen as a response to specific previous findings [16]. There is some evidence to suggest that there may be a correlation between the WAI and age and working period. Some studies have found that older workers and those with longer working periods tend to have lower WAI scores, while younger workers and those with shorter working periods tend to have higher WAI scores [34]. This may be due to the increased physical and mental demands of work as well as the accumulation of health problems over time.

The purpose of occupational health management that may be achieved by implementing WAI is to consider maintaining and making efforts to improve the quality of work for an inclusive society that includes people with disabilities in accordance with the UN’s Sustainable Development Goals (SDGs). It has been revealed that 54.2% (n = 84) of the total number of disabled workers who participated in the RTW program significantly improved their working abilities due to their participation. During the COVID-19 outbreak in Indonesia, it was observed that 27.3% (n = 42) of employees fell into the moderate group, while 18.2% (n = 28) had an exceptionally excellent work ability index.

## 5. Conclusions

In conclusion, our study found that the return to work program was effective in improving the quality of life and work ability of disabled workers during the COVID-19 pandemic, as seen by the significant difference in scores between the RTW and non-RTW participants. Policy-makers and employers should consider using assessment tools and gathering feedback to regularly assess the program and make necessary improvements.

### Limitations and Strengths

There were some flaws in how the study was conducted that need to be pointed out. It is important to consider the limitations of our study including the sample size and specific geographic location. Further research with a larger sample size and diverse location is needed to further understand the effectiveness of RTW programs for disabled workers during the COVID-19 pandemic. The data collection was conducted using a purposive sample, which refers to situations during the pandemic that allowed researchers to approach the subjects through indirect encounters or online, which profoundly affected the willingness of participants to continue with the study, resulting in possible biases. There are, nevertheless, certain advantages to consider. We demonstrated the importance of contextual factors in implementing RTW, especially when examining the program’s outcomes in measuring the functional ability and quality of life of workers with disabilities. This was carried out by using various data from very different social and cultural backgrounds and a wide variety of social assistance background frameworks.

## Figures and Tables

**Figure 1 ijerph-20-03094-f001:**
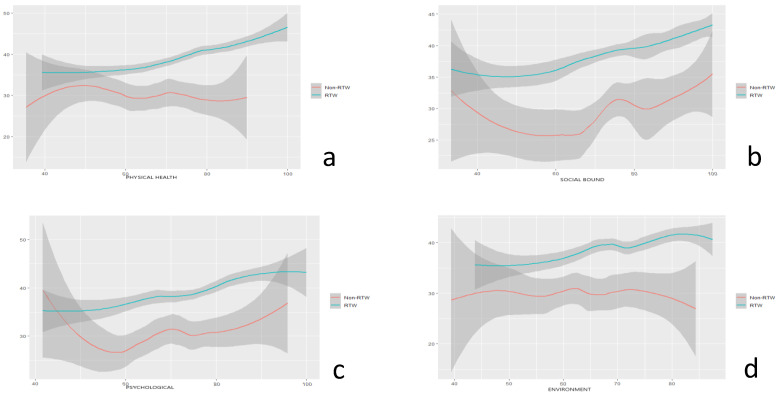
The comparative value of the domains of the disabled employees’ QoL in relation to WAI among the RTW and those who did not participate in the RTW. (**a**) Physical health, (**b**) social bound, (**c**) psychological, and (**d**) environmental health.

**Figure 2 ijerph-20-03094-f002:**
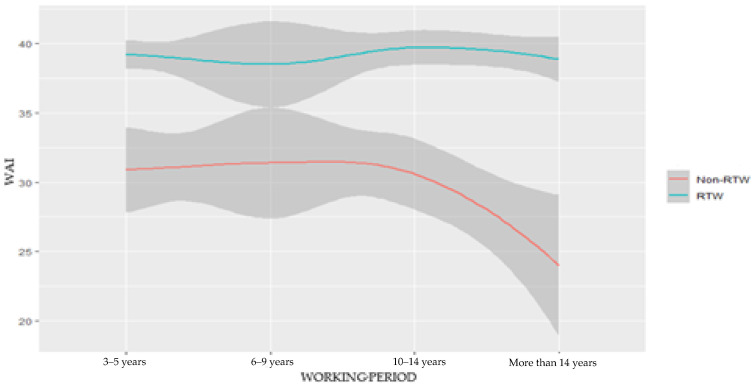
Distribution score of the work ability index based on the age and working period.

**Figure 3 ijerph-20-03094-f003:**
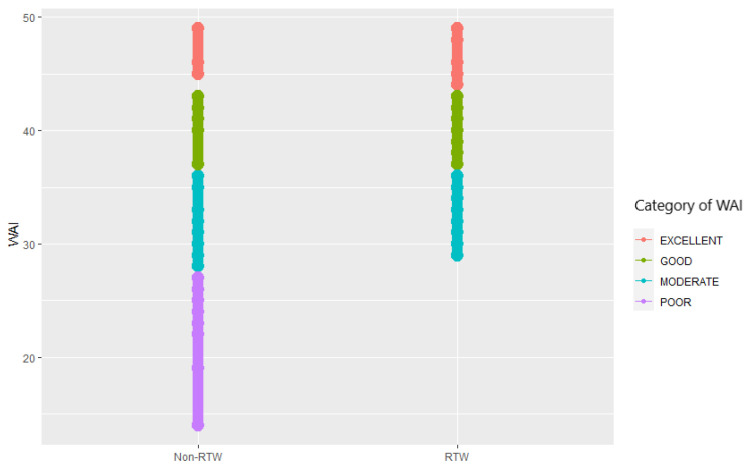
Work ability index dynamic value based on the category.

**Table 1 ijerph-20-03094-t001:** Statistic result comparison of RTW and non-RTW participants.

Variables		RTW	Non-RTW	Mann–Whitney, *t,* and χ^2^	Multivariate Logistic Regression	
		N (%) or Mean ± SD	N (%) or Mean ± SD	*p*-Value	OR (95% CI)	*p*-Value
Age	69(30.43%)	39(16.96%)	39 ± 16.96%	0.430	3.726	0.847
	77(33.48%)	35(15.22%)	35 ± 15.22%			
	8(3.48%)	1(0.43%)	1 ± 0.43%			
Gender	33(14.40%)	24(10.50%)	24 ± 10.50%	0.083	0.184	0.125
	121(52.80%)	51(22.30%))	51 ± 22.30%			
Marital status	24(10.50%)	13(5.70%)	13 ± 5.70%	0.736	4.876	0.585
	130(56.80%)	62(27.10%)	62 ± 27.10%			
Job description	146(63.80%)	72(31.40%)	72 ± 31.40%	0.692	2.792	0.769
	8(3.50%)	3(1.30%)	3 ± 1.30%			
Level of education	132(57.60%)	64(27.90%)	64 ± 27.90%	0.939	0.283	0.367
	22(9.60%)	11(4.80%)	11 ± 4.80%			
Work Period	8(3.50%)	12(5.20%)	12 ± 5.20%	0.000 *	0.430	0.039 *
	48(21.00%)	33(14.40%)	33 ± 14.40%			
	69(30.10%)	22(9.60%)	22 ± 9.60%			
	29(12.70%)	8(3.50%)	8 ± 3.50%			
Location of residence	83(36.52%)	36(15.65%)	36 ± 15.65%	0.493	1.143	0.264
	10(4.35%)	7(3.04%)	7 ± 3.04%			
	61(26.52%)	32(13.91%)	32 ± 13.91%			
Work Ability Index	0(0.00%)	37(16.20%)	37 ± 16.20%	0.000 *	0.692	0.000 *
	42(18.30%)	20(8.70%)	20 ± 8.70%			
	84(36.70%)	14(6.10%)	14 ± 6.10%			
	28(12.20%)	4(1.70%)	4 ± 1.70%			
Quality of Life	72.94 ± 11.94	65.91 ± 10.72	65.91 ± 10.72	0.000 *	0.988	0.374
	74.35 ± 10.13	70.84 ± 9.82	70.84 ± 9.82	0.016 *	0.956	0.367
	76.14 ± 14.45	74.13 ± 14.17	74.13 ± 14.17	0.388	1.029	0.055
	69.89 ± 8.78	63.64 ± 8.52	63.64 ± 8.52	0.000 *	1.062	0.023 *

* *p* < 0.05 indicates a significant difference between the two groups (RTW and Non-RTW) at a level of 95% confidence. * *p* < 0.01 indicates a highly significant difference between the two groups at a level of 99% confidence.

## Data Availability

The authors state that all data validated in the report’s findings are included in the document.

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
