# Peer review of "Analysis of the Return to Work Program for Disabled Workers during the Pandemic COVID-19 Using the Quality of Life and Work Ability Index: Cross-Sectional Study"

_ijerph, 2023, doi:10.3390/ijerph20043094_

Round 1

Reviewer 1 Report (Previous Reviewer 2)

I appreciate the work that the authors have done in revising this study, and providing more information about the sample. However, I think the authors' missed my point a little -- I wasn't concerned about sample size. I was concerned about selection bias. That is, what unobserved characteristics and programmatic experiences that the authors did not have information on what influenced which people selected themselves into the sample.

The paper has been improved significantly, but I wish the potential for a positive program effect bias was highlighted a bit more.

Author Response

Dear Reviewer,

I hope this email finds you well. I am writing to express my gratitude for taking the time to review our manuscript, "Analysis of the Return to Work Program for Disability Workers During the Pandemic Covid-19 Using the Quality of Life And Work Ability Index: Cross-Sectional Study," and to provide such insightful feedback.

We appreciate your detailed and thoughtful comments and have taken them into consideration in revising the manuscript. I am pleased to attach a response to your comments, which addresses each of the points you raised and outlines the changes we have made to the manuscript in response to your feedback.

Your comments have helped us to significantly improve the quality and clarity of our study, and we are grateful for the opportunity to incorporate your suggestions. We hope that the revisions have fully addressed your concerns and that you will find the revised manuscript to be of a high standard.

Thank you again for your time and effort in reviewing our work. We truly appreciate the opportunity to benefit from your expertise and knowledge.

Best regards,
Arie Arizandi Kurninato, Pharm.D., M.OHS.

Reviewer 2 Report (New Reviewer)

Thanks for inviting me to review this manuscript. This is a very interesting, well-organised and insightful paper. I have only one major comment:

I would like to see additional discussions about: (a) other policy instruments that can facilitate disadvantaged workers’ work resumption during Covid (see Li et al., 2022; Zamani et al., 2022). Also, please see Gunn et al. (2022) for a review of initiatives dealing with precarious employment. And (b) policies that influenced disadvantaged people such as disability workers’ return to work during the pandemic. For example, Liu et al. (2022) discussed that Chinese migrant workers’ work resumption may be influenced by discriminatory Covid containment policies.

Reference

Gunn, V., Kreshpaj, B., Matilla-Santander, N., Vignola, E. F., Wegman, D. H., Hogstedt, C., ... & Håkansta, C. (2022). Initiatives Addressing Precarious Employment and Its Effects on Workers’ Health and Well-Being: A Systematic Review. International Journal of Environmental Research and Public Health, 19(4), 2232.

Li, X., Hui, E. C., & Shen, J. (2022). Institutional development and the government response to COVID-19 in China. Habitat International, 127, 102629.

Liu, Q., Liu, Z., Kang, T., Zhu, L., & Zhao, P. (2022). Transport inequities through the lens of environmental racism: rural-urban migrants under Covid-19. Transport policy, 122, 26-38.

Zamani, S. H., Rahman, R. A., Fauzi, M. A., & Yusof, L. M. (2022). Government pandemic response strategies for AEC enterprises: Lessons from COVID-19. Journal of Engineering, Design and Technology.

Author Response

Dear Reviewer,

I hope this email finds you well. I am writing to express my gratitude for taking the time to review our manuscript, "Analysis of the Return to Work Program for Disability Workers During the Pandemic Covid-19 Using the Quality of Life And Work Ability Index: Cross-Sectional Study," and to provide such insightful feedback.

We appreciate your detailed and thoughtful comments and have taken them into consideration in revising the manuscript. I am pleased to attach a response to your comments, which addresses each of the points you raised and outlines the changes we have made to the manuscript in response to your feedback.

Your comments have helped us to significantly improve the quality and clarity of our study, and we are grateful for the opportunity to incorporate your suggestions. We hope that the revisions have fully addressed your concerns and that you will find the revised manuscript to be of a high standard.

Thank you again for your time and effort in reviewing our work. We truly appreciate the opportunity to benefit from your expertise and knowledge.

Best regards,
Arie Arizandi Kurninato, Pharm.D., M.OHS. PhD 

Reviewer 3 Report (New Reviewer)

General Comments

1.      This paper presents an interesting comparison between workers who do and do not participate in a return-to-work program in Indonesia.

2.       Unfortunately, the paper is poorly motivated and written, making it hard to follow. Among other things, I suggest authors have the manuscript professionally edited before their next submission.

Abstract

1.       P1, row 20: remove the second spelling-out of “return-to-work”

2.       P1, row 21: introduce the “WAI” acronym after first mention (and then use it throughout the abstract)

3.       P1, row 25: spell out WHOQoL-BREF

4.       P1, rows 25-28: I could not follow the sentence—please revise for clarity and include actual differences, not just p-values of differences.

Introduction

1.       The first paragraph is virtually unreadable, containing both grammatical and logical errors. Below are just two examples of these problems:

a.       P1, rows 36-37. “Occupational accident insurance” is a program that in most countries has existed for decades—not just enacted in response to COVID-19. Authors should indicate whether governments made a change to their existing accident insurance programs in response to COVID-19.

b.       P1, rows 36-38. “For this reason” is not appropriate to begin this sentence, because the previous sentence does not include the relevant reason.

2.       P2, rows 60-64: Authors should clarify why the Indonesian program is so unique compared to other developing nations. For example, if it’s the only developing nation offering an RTW program in it’s social security system, then say that rather than “In contrast to other developing nations”.

3.       Other paragraphs in the introduction are equally problematic in terms of grammar and logic. There is also a lack of relevant citations (for example, about disability discrimination and participation in the labor market).

Methods

1.       Before the methods section, Authors should include a section describing the RTW program and who is eligible to participate.

2.       The methods section should clarify the difference between the RTW and nonRTW groups—how is each group defined?

3.       Page 3 rows 134-137: please clarify if the case managers where originally employed by BPJS or were added specifically for the current study.

4.       P3, rows 147-152: Authors should clarify why it’s important to “check for normality in the data”, what are “free variables”, and what they mean by saying “r was so little that it didn’t need a definition”.

5.       P3, rows 153-156: The described logistic regression cannot evaluate any causal relationship—only associations. Instead of mentioning causality, clarify you are assessing associations while controlling for other factors. It would also help to clarify which relationships are of primary interest.

Results

1.       P4, rows 160-161. Unclear how this research can “explore the disruptive effects of COVID-19” when there is no comparison to a different time period before COVID-19.

2.       P4, rows 167-175. I suggest moving the methods-related information (on tests and regressions) into the methods section above and focusing here on the results in the tables.

3.       Table 1: There are many problems here. For example: I do not understand what the numbers in the RTW and non-RTW stand for. How can the age group (and all other) percentages add up to more than 100? What are the sample sizes? What is the dependent variable for the multivariate logistic regression? What relationship is the OR estimate capturing? Where is the CI for the OR estimate?

4.       P5, rows 179-187: the paragraph following Table 1 should discuss results shown in the table.

5.       P5, rows 193-194: the final sentence in paragraph is incomplete.

6.       Figure 1: clarify what is on horizontal axis and what is on vertical axis.

7.       P5, rows 223-224: clarify what this means: “the quality of life for disabled workers is substantial”. In comparison to what?

8.       Either here, in the conclusion, or both, authors should clarify that the difference in WAI and QOL between the RTW and non-RTW group might be due to healthier people selecting the RTW option.

9.       Figure 2: unclear how the horizontal axis can capture both age and work period. Also unclear what “work period” means.

10.   P7, rows 280-282: I could not understand this sentence. Please revise to make it clear.

11.   Page 7, rows 283-290: Please revise so as not to attribute the differences between RTW and nonRTW as causal. For example, one could speculate that more workers with “poor” WAI did not believe they could benefit from RTW services.

 Discussion

1.       P7, rows 293-328: Authors should avoid attributing difference to causality, for reasons mentioned above. Instead, I suggest discussions the associations they found and saying these could be due to a combination of impacts of and selection into the RTW program.

2.        P7, rows 330-332: It is inaccurate to claim this study contributes to the literature on the impact of COVID-19, because no comparisons where made to a period without COVID-19.

3.       As in the introduction, much of the text here is problematic in terms of grammar and logic.

 Conclusions

1.       See some of the comments above, that apply here too.

2.       Authors should add the inability to attribute causality as a limitation.

Author Response

Dear Reviewer,

I hope this email finds you well. I am writing to express my gratitude for taking the time to review our manuscript, "Analysis of the Return to Work Program for Disability Workers During the Pandemic Covid-19 Using the Quality of Life And Work Ability Index: Cross-Sectional Study," and to provide such insightful feedback.

We appreciate your detailed and thoughtful comments and have taken them into consideration in revising the manuscript. I am pleased to attach a response to your comments, which addresses each of the points you raised and outlines the changes we have made to the manuscript in response to your feedback.

Your comments have helped us to significantly improve the quality and clarity of our study, and we are grateful for the opportunity to incorporate your suggestions. We hope that the revisions have fully addressed your concerns and that you will find the revised manuscript to be of a high standard.

Thank you again for your time and effort in reviewing our work. We truly appreciate the opportunity to benefit from your expertise and knowledge.

Best regards, Arie Arizandi Kurninato, Pharm.D., M.OHS. 

Round 2

Reviewer 2 Report (New Reviewer)

Thanks for revising. The discussion section reads much better now. The authors have sufficiently dealt with my first comment but it seems the authors forgot to deal with the second one.

Maybe I did not make myself clear enough last time. In my second comment, I’m saying that disabled people are one particular type of disadvantaged population group. So it is interesting to put disabled people within a wider range of disadvantaged people, such as rural-urban migrants in China (I mentioned in the previous review report). How were they influenced differently by the pandemic? Would some instruments be very helpful for some disadvantaged people but not very helpful to others? What would the findings inform the policymaking that aims to promote the work resumption of other disadvantaged people?

Author Response

Thank you for your feedback. We apologize for missing your second comment in the previous revision. We understand that disabled people are one type of disadvantaged population group and recognize the importance of considering the experiences of other disadvantaged groups, such as rural-urban migrants in China. We have now added a section in the discussion to include relevant studies and discussions on these groups, and how they may be influenced differently by events such as the pandemic. Thank you for your insights and for helping us to further improve our research.

Reviewer 3 Report (New Reviewer)

  • Thanks for the opportunity to take another look at this interesting paper. I appreciate the authors addressing most of my comments. I have two remaining substantive comments below.
  • Rows 234-240 (and other places): I maintain that it is incorrect to claim that the research "aimed to understand the impact of the pandemic on the results of the RTW program for employees with impairments." The study compares outcomes for RTW participants and non-participants. While the pandemic might have affected the program and outcomes, the study is not evaluating those affects. You can say that you compared outcomes for RTW and non-RTW participant during the pandemic, without saying you evaluated the effects of the pandemic.
  • You are still attributing your findings to causality, which remains inappropriate. For example, in rows 336-338 (and other places) this text still implies causality "the RTW program is effective in enhancing the work ability of disabled workers". You can say that the RTW program is associated with higher WAI, but need to be clear that the study limitations are such that you cannot attribute causality. Again, I suggest you consider the possibility that those who choose to participate in the RTW program might indeed be "healthier" or are on average more motivated to work than those who choose not to. "Healthier" doesn't mean that they are not injured. It means their injury is less severe, but in a way that is not captured in the data (for example, if you measured their WAI at the beginning of the program, it very well could be that fewer in the group had a "poor" WAI even before the program started). This is not to say that the RTW program's impact is not positive, just that you need a different sort of analysis (like a randomized experiment) to attribute causality)

Author Response

Thank you for your continued feedback. We have revised the language in the paper to better reflect the study's comparison of RTW participants and non-participants during the COVID-19 pandemic. We acknowledge that the pandemic may have impacted the program and outcomes, but the study is not evaluating these effects. Furthermore, we have revised the text to accurately reflect the limitations of the study, acknowledging that a randomized experiment would be needed to attribute causality to the RTW program's impact on enhancing work ability of disabled workers. We appreciate your suggestion to consider the possibility of differences in severity of impairments and motivation to work between RTW participants and non-participants.

This manuscript is a resubmission of an earlier submission. The following is a list of the peer review reports and author responses from that submission.

Round 1

Reviewer 1 Report

Thank you for the invitation to review this manuscript. The English expressions and level of detail in this manuscript is extremely poor. I could not read beyond the first paragarph of the Results. I hope the feedback provided can assist the authors. 

Introduction:

·       The sentence on line 37 – 38 is grammatically incorrect

·       the sentence on line 42 – 43 does not make sense

·       the sentence on line lines 46 – 48 does not make sense:

“Workers with disabilities, who are also performers in the labor market, have also been vulnerable to the effect (of what?) particularly in the course of performing their routine daily duties.” The link between covid-19 and workers with disability performing daily duties is not clear

The Return to Work program needs to be more clearly described. What is meant by the “in-kind benefit services”? Who provides these services? Which workers are able to utilise the Return to Work program? is it only those with pre-existing disabilities or those who sustain a work place injury? What injuries are covered by the workers compensation scheme?

Line  56: what ‘operation’? I suspect this does not refer to surgical operation

Line 84: the study aims suggest this paper is a case study. Please correct

Data collection

A statement about ethical approval is needed and the name of the approving instituition.

If 93% of the invited sample agreed to participate this is not the ‘whole population’

Data analysis

A section on data analysis MUST be included

Results

This sentence is inconsistent with the aims of the research “The research was undertaken during the outbreak of COVID-19 to explore the disruptive effects of COVID-19 on RTW results in terms of the quality of life and job capacity of impaired employees.”

Apologies, i could not continue reading as it was very difficult to follow

Author Response

Dear Reviewers,

Thank you for your thoughtful comments on our article. We have carefully considered your feedback and have made several changes to the article in response to your suggestions. We believe these changes improve the clarity and rigor of the study, and we hope they address your concerns. Please find included our responses to your past review.

We have also made several changes to address the grammatical errors that you pointed out. We have revised the language throughout the article to ensure that it is clear and easy to understand. We hope these changes have improved the overall quality of the article.

Thank you again for your valuable feedback. We appreciate your efforts to help us improve the article.

Sincerely,

Authors

Reviewer 2 Report

My main issue is with the sample design. I believe it undermines the ability to draw the conclusions the authors draw. The sample only consists of people who (a) participated in a RTW program and (b) volunteered to be in the sample. Both of these things make the sample non-random.

The best case scenario would be to randomly select workers who have had injuries. Then we could compare those not involved in RTW programs versus those who did. Without the people not participating in RTW programs we can draw no conclusions about the effect of those programs.

Moreover, these are people who volunteered to be in the sample. This makes them different, on average, from the people who were contacted but who did not join the sample. Maybe the latter people have a very low QoL, had a bad experience with RTW programs, and want nothing to do with them -- even in a study. Who knows? That may not be the case, but the fact that they refused to participate undermines the usefulness of the sample.

At the very least the authors should compare the people who agreed to be in the sample with those who didn't. Are they more likely to be men? Older? From certain industries? Certain regions?  If those not participating looked like those who did, that would be a bit better -- but still there could be non-observable characteristics (or experiences) that led some people to agree to be in the sample and some not.

There are methods for accounting for selection bias in samples (see Heckman-Singer) but that requires information on everyone, including some variable that might affect participation but not the impact of the RTW program.

Finally, how can we say anything about the impact of RTW on ability to work without knowing what the ability to work score was prior to entering the program?

Also, maybe because of English not being the authors' primary language there were some things I found a bit confusing.

Line 169-  why isn't psychological mentioned? It has a high summary QoL score than physical?

Line 183 - You can't say the quality of life is substantial without looking at non-disabled people. On what basis is a score of 75 good? What if it is 95 in the general population?

Line 186 - they state the minimum score of socially bound is 33.3 in Table 2, but Table 2 has correlation coefficients? I'm not sure what they are referring to.

Line 188 - What does it mean to say that only one out of 154 have something lower than average? How can 153 out of 154 people be above average? What average are you talking about? I think this is probably an English issue.

Author Response

Dear Reviewer,

Thank you for your detailed comments on our article, "Analysis of the Return to Work Program for Disability Workers During the Pandemic Covid-19 Using the Quality of Life And Work Ability Index: Cross-Sectional Study." We have carefully considered your feedback and have made several changes to the article in response. We believe that these changes have greatly improved the clarity and logical flow of the article. Please find included our responses to your past review.

We have also addressed the grammatical errors pointed out by the reviewer. We have revised the language throughout the article to ensure that it is clear and easy to understand for your readers. We have attached a marked-up version of the article showing the changes we have made.

Thank you again for your valuable feedback. We hope that these changes address your concerns and that the revised article meets your approval.
